METHODS

# COEXIST: Coordinated single-cell integration of serial multiplexed tissue images

Robert T. Heussner[1☉], Cameron F. Watson[1,2☉], Christopher Z. Eddy[3], Kunlun Wang[1], Eric M. Cramer[1], Allison L. Creason[1,2], Gordon B. Mills[2], Young Hwan Chang [1,2*]

**1** Department of Biomedical Engineering and Computational Biology Program, Oregon Health & Science University (OHSU), Portland, Oregon, United States of America, **2** Knight Cancer Institute, Oregon Health & Science University, Portland, Oregon, United States of America, **3** Cancer Early Detection Advanced Research (CEDAR), Knight Cancer Institute, OHSU, Portland, Oregon, United States of America

☉ These authors contributed equally to this work.
* chanyo@ohsu.edu

## Abstract

Multiplexed tissue imaging (MTI) and other spatial profiling technologies commonly utilize serial tissue sectioning to comprehensively profile samples by imaging each section with unique biomarker panels or assays. The dependence on serial sections is attributed to technological limitations of MTI panel size or incompatible multi-assay protocols. Although image registration can align serially sectioned MTIs, integration at the single-cell level poses a challenge due to inherent biological heterogeneity. Existing computational methods overlook both cell population heterogeneity across modalities and spatial information, which are critical for effectively completing this task. To address this problem, we first use Monte-Carlo simulations to estimate the overlap between serial 5μm-thick sections. We then introduce COEXIST, a novel algorithm that synergistically combines shared molecular profiles with spatial information to seamlessly integrate serial sections at the single-cell level. We demonstrate COEXIST necessity and performance across several applications. These include combining MTI panels for improved spatial single-cell profiling, rectification of mis-called cell phenotypes using a single MTI panel, and the comparison of MTI platforms at single-cell resolution. COEXIST not only elevates MTI platform validation but also overcomes the constraints of MTI's panel size and the limitation of full nuclei on a single slide, capturing more intact nuclei in consecutive sections and thus enabling deeper profiling of cell lineages and functional states.

## Author summary

Multiplex tissue imaging (MTI) allows researchers to study tissue samples by measuring many proteins in individual cells while preserving their spatial organization, providing critical insights into the tumor microenvironment — the complex

**Data availability statement:** Data Availability As part of this paper all images at full resolution, all derived image data (e.g. segmentation masks), and all cell count tables are publicly released via the NCI-recognized repository for Human Tumor Atlas Network (HTAN; https://humantumoratlas.org/) at Sage Synapse (associated Identifiers: HTAN TNP – TMA, OHSU_TMA1_004-XX, https://www.synapse.org/Synapse:syn17022193/wiki/596115) where XX represents TMA core ID. The tonsil image dataset used for the label propagation experiment is available at https://www.synapse.org/MCMICRO_images and are available through the Human Tumor Atlas Network. The CRC WSI dataset is available through the Human Tumor Atlas Network (associated identifiers: HTAN TNP – SARDANA, HTA13_1_6, https://www.synapse.org/Synapse:syn18434611/wiki/590458). To request access, please log in or create a free Synapse account at https://www.synapse.org, and follow the procedures for data access as described on the respective dataset page. Code Availability All software used in this manuscript is detailed in the article's Methods section and its Supplemental Information. The associated scripts are freely available via GitHub as described at https://github.com/heussner/coexist.

**Funding:** This work was carried out with major support from National Cancer Institute (NCI) Human Tumor Atlas Network (HTAN) Research Centers at OHSU (U2CCA233280 to G.B.M and Y.H.C). Y.H.C is supported by R01 CA253860, U01 294548 and Kuni Foundation Imagination Grants. The resources of the Exacloud high-performance computing environment developed jointly by OHSU and Intel and the technical support of the OHSU Advanced Computing Center are gratefully acknowledged. The funders had no role in study design, data collection and analysis, decision to publish, or preparation of the manuscript.

**Competing interests:** I have read the journal's policy and the authors of this manuscript have the following competing interests. The authors declare the following competing interests: G.B.M. is a SAB member or Consultant: for Amphista, Astex, AstraZeneca, BlueDot, Chrysallis Biotechnology, Ellipses Pharma, GSK, ImmunoMET, Infinity, Ionis, Leapfrog Bio, Lilly, Medacorp, Nanostring, Nuvectis,

system of cells and structures surrounding tumors that influences cancer progression and treatment. However, current MTI platforms are limited in the number of biomarkers that can be stained simultaneously on a single tissue section due to panel size restrictions or assay incompatibilities. To expand the number of proteins measured, researchers often apply different panels to consecutive thin tissue sections. While image registration can align these sections, integrating them at the single-cell level remains challenging due to biological heterogeneity and differences in cell composition across slices. Existing methods often fail to fully incorporate both molecular and spatial information. To address this, we developed COEXIST, a computational algorithm that combines shared molecular profiles with spatial information to match cells across serial sections, enabling more complete and accurate tissue profiling. COEXIST improves MTI utility, enhances single-cell resolution, and offers opportunities for integrating MTI with spatial transcriptomics for deeper biological insight.

## Introduction

Multiplexed tissue imaging (MTI) provides spatially resolved molecular profiling of single cells in thin tissue sections [1]. Validating MTI platforms and panels is essential for reproducibility [2], and is often achieved by comparing their performances on serial tissue sections. This comparison assumes uniformity across the sections and is conducted either at the pixel level by marker intensity or at the population level by cell phenotyping [3–6]. However, tissue sections are inherently heterogeneous, driven by factors including the sectioning process, the complexity of the tissue structure, or the presence of rare cell populations [7–10]. A recent study applying cyclic immunofluorescence imaging (CyCIF) to 40µm-thick melanoma sections found more than 90% of the nuclei were not fully intact within 5µm subsections [11]. This finding challenges the prevailing homogeneity assumption in bulk-level cell populations across serial sections. Another study of 4µm thick histological tissue sections showed their mean pixel-wise Pearson correlation was less than 0.6, underscoring their biological heterogeneity [12]. Moreover, fundamental challenges like spectral overlap and tissue degradation limit MTI panels to 40–100 proteins per sample [1]. By contrast, non-spatial technologies like CITE-seq and scRNA-seq can measure over 200 epitopes [13] and 1000 unique molecular identifiers [14] per cell, respectively. Although applying different MTI panels or spatial assays on serial sections allows for deeper molecular profiling, it sacrifices paired single-cell measurements. A computational approach to harmonize these datasets at the single-cell level would mitigate the challenge of unpaired single-cell measurements, improve rigorous validation of MTI technologies, and overcome MTI's panel size limitations [15–17].

Early methods for tracking nuclei across serial electron microscopy images rely on manually drawn contours or thresholding to track objects across sections [18]. While modern image registration algorithms can align serial MTIs by identifying shared landmarks [19–22], achieving perfect alignment at the single-cell level is impossible

PDX Pharmaceuticals, Qureator, Roche, Signalchem Lifesciences, Tarveda, Turbine, Zentalis Pharmaceuticals. G.B.M. has Stock/ Options/Financial relationships with: Bluedot, Catena Pharmaceuticals, ImmunoMet, Nuvectis, SignalChem, Tarveda, and Turbine. G.B.M. has Licensed Technology: HRD assay to Myriad Genetics, DSP patents with Nanostring. G.B.M. has Sponsored research with AstraZeneca. The other authors declare no competing interests.

due to inherent biological variation (Fig 1A). These types of approaches either require exceptionally thin sections, do not incorporate the full molecular profiles of the cells, or do not perform unified single-cell level analysis [8,20,22,23]. Furthermore, our previous work reported natural variations between serial sections in the image-to-image translation task, especially when using consecutive H&E sections to infer immunofluorescence intensity [19,21]. Recent platforms such as RareCyte Orion [16] or studies [24] are prioritizing multimodal assays such as CyCIF and hematoxylin and eosin (H&E) or high-plex protein combined with whole transcriptome sequencing in the same section to address this problem. Beyond MTI, many spatially resolved single-cell assays utilize serial tissue sections due to technological constraints [15], including Nanostring CosMX [25], spatial proteogenomics [26], and others [27].

Alternatively, computational multi-omic integration methods aim to pair single-cell datasets and can be extended to spatial assays [28]. These methods assume similar cell states across datasets and use techniques such as canonical correlation analysis (CCA) [29] or variational autoencoders [30] to jointly embed single-cell representations and match them across modalities [31]. Linear sum assignment (LSA) [32,33] is frequently used as a framework to achieve this matching by constructing a cost matrix of potential cell pairs and identifying a set of pairs of minimal overall cost [34,35]. For example, MaxFuse [36], a recent approach backboned by LSA, is specifically designed for weakly linked modalities with few shared features, including proteomic, transcriptomic, and epigenomic information at single-cell resolution on the same tissue section. It generates nearest neighbor graphs, identifies anchor cells based on linked feature correlation, and then iteratively refines matches with graph smoothing and CCA. However, current multi-omic integration strategies are not compatible with spatial assays, as they fail to include spatial information and incorrectly assume homogeneity within the population, particularly when dealing with serial sections. Alternatively, cell tracking algorithms [37], designed to track cell movements across video frames, can be adapted to treat serial images as sequential frames but do not leverage the rich molecular profiles provided by MTI. Three-dimensional cell segmentation methods, such as the 3D extension of the Cellpose algorithm, similarly do not consider molecular profiles of cells across adjacent sections, are limited by the need for very thin tissue sections (2–3μm) and impose strict overlap thresholds to consider a cell tracked [38]. Therefore, it is essential to develop a single-cell integration method that combines molecular characteristics with spatial context to effectively integrate multi-modal or multi-panel datasets obtained from serial tissue sections.

To address this, we first design a Monte Carlo simulation to estimate the theoretical overlap of single cells across consecutive tissue sections (Fig 1B). We then examine how LSA can be used with cell tracking algorithms, highlighting the importance of an approach based on tracking. Next, we introduce COEXIST, which leverages spatial information and shared biomarker expression to pair individual cells across consecutive tissue sections (Fig 1C). We apply our method to the Human Tumor Atlas Network (HTAN) Trans-Network Project tissue microarray (TNP-TMA) breast cancer (BC) dataset, where we phenotype cells using a combined tumor and immune panel of 42 unique molecular features, nearly doubling the number of

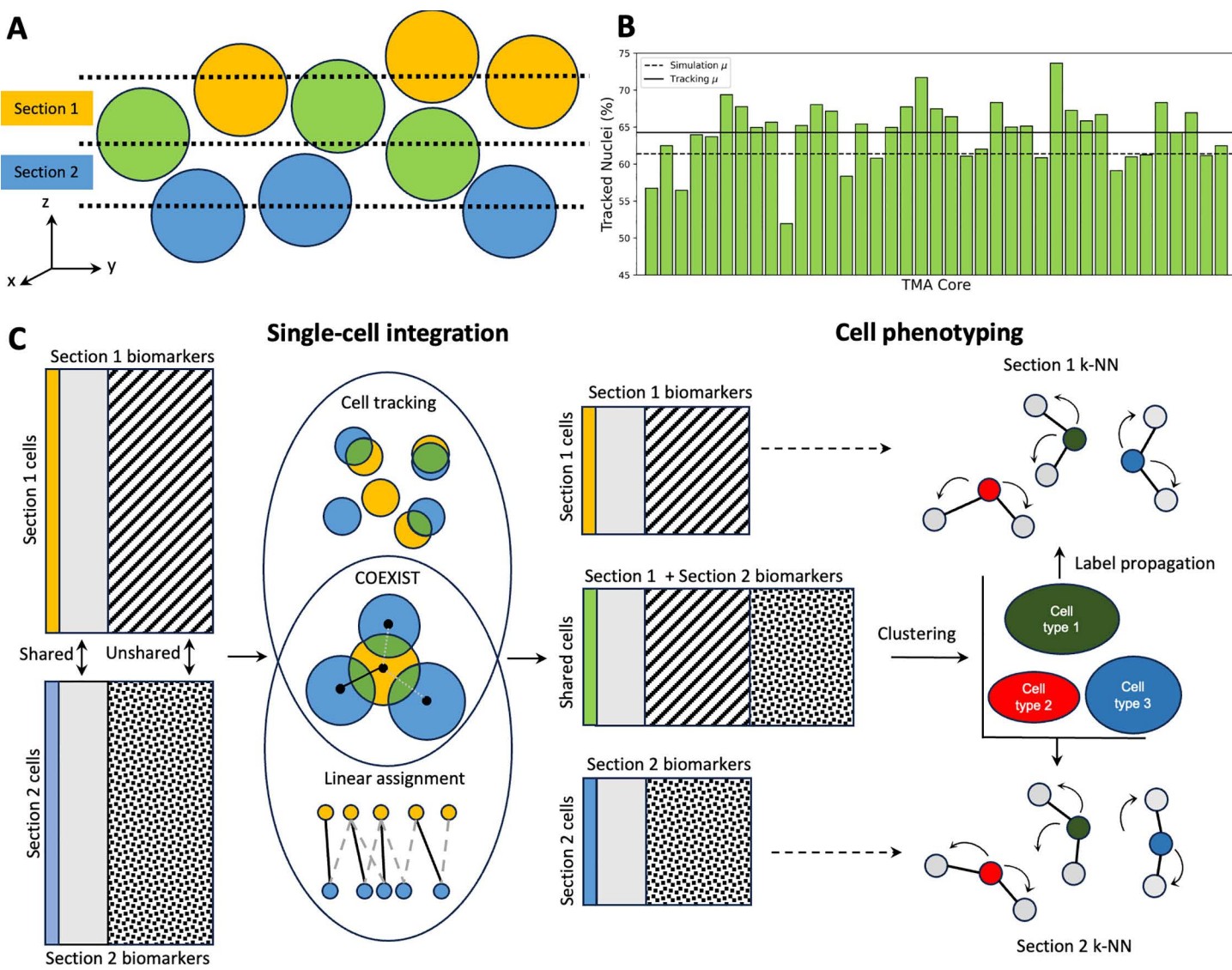

**Fig 1. COEXIST integrates serial multiplexed tissue images. (A)** Visualization of shared (green) and unshared (yellow/blue) cells across serial tissue sections. **(B)** Monte-Carlo simulation estimates approximately 62% of cells are present in serial sections. **(C)** Overview of single-cell integration and downstream analysis: Cell-feature tables of serial tissue sections with shared and unshared features are integrated with an algorithm that accounts for the spatial and protein expression information, identifying the shared (green) and unshared (yellow, blue) cell populations. The shared cells undergo phenotyping using the combined panel, with resulting cell type labels being extended to the remaining unshared cells for each section. This propagation of cell type labels to the unshared cells is achieved via k-NN.

features compared to a single panel. Our algorithm allows for the full reconstruction and characterization of 2.5D tissue, making it highly versatile for integrating spatial multi-omics data to analyze emerging spatial assays [39].

## Results

### Challenges in spatial profiling across serial sections

Our Monte Carlo simulation (**Methods**) indicates that approximately 62% of cells are shared across consecutive sections in the TNP-TMA BC dataset, which employs 5μm-thick sections and has a mean nuclear diameter of 7.8μm (with

a standard deviation of 2.4µm). To simplify the simulation, we assumed nuclei to be spheres with normally distributed diameters, drawing nuclear size statistics directly from each image. At each simulation iteration, the nucleus diameter was sampled from a normal distribution parameterized by the observed nuclear diameter distribution (mean 7.8µm, SD 2.4µm) across all tissue cores, thereby accounting for the prevalence of smaller nuclei in the tissue. While the purpose of the simulation was to provide a rough estimate of the fraction of nuclei shared across consecutive sections, it aligned with recent literature [11] and closely matched results from the cell tracking algorithm. Additionally, we repeated the simulation using the nuclear size statistics and section thickness from another study of 3D tissue CyCIF [11]. Our simulation yielded comparable results to the ground-truth measurements: 89.3% of nuclei were not fully intact in 5µm thick sections (compared to >90%) [11], while 71.3-77.6% remained fully intact within 30–40µm sections (compared to 60–80%) [11]. Thus, our simulation provides a reliable estimate for the fraction of cells shared across serial sections, highlighting that both nuclear size and section thickness independently affect the probability of a nucleus appearing in multiple sections. The degree of cell colocalization varies substantially with section thickness and cell size, as shown in Fig 1B, where each bar represents a different TMA core. These results motivate a new approach to allow rigorous validation and integration of serial sections at the single-cell level.

## Improved integration using cell tracking over linear sum assignment (LSA)

Many multi-omics integration methods operate with the assumption that cell populations are shared across serial sections, allowing for the matching of populations by linking cells within the same molecular states. However, our simulation revealed applying this assumption directly to spatial assays and solely focusing on optimizing linked molecular feature correlation via LSA, such as in MaxFuse [36], does not guarantee spatial coherence of the matches (S1A Fig). Alternatively, adopting cell tracking algorithms that rely on cell morphology and location for this problem can guarantee spatial coherence [37] (S1B Fig). In our subsequent analysis, we investigate the use of both approaches for the integration problem and propose a novel algorithm that overcomes their respective limitations. We frame the problem as finding a single-cell matching between a reference and target image, representing consecutive MTIs. We define match quality as the mean Spearman correlation of the shared molecular features of the paired cells.

LSA achieves higher match quality compared to cell tracking given the same reference cells. This is because LSA directly optimizes shared feature correlation, whereas cell tracking does not, and considers matches across arbitrary distances, while cell tracking cannot (S1A Fig). Therefore, to implement LSA we introduced spatial constraints to the Jonker-Volgenant (JV) [40] algorithm to restrict potential matchings to a local neighborhood of size $r$. This modification also mitigates local registration discrepancies, overcoming possible failures in image registration. To create a fair benchmark for LSA and cell tracking, we first identified $N$ anchor cells from a reference MTI that comprised $M$ ($M > N$) cells using the cell tracking algorithm. This ensured that the benchmark reference cells were associated with spatially coherent target candidates (**Methods**). We then used our spatially constrained LSA implementation to identify target matches for the $N$ anchors, iteratively expanding the radius parameter. Additionally, we conducted LSA by randomly sampling $N$ anchor cells from the reference slide to test the importance of cell tracking in choosing suitable anchor cells instead of using random anchor cells. We then repeated the radial expansion process to create sets of matches.

Within neighborhoods smaller than 50µm, or approximately 6 nuclei, where the true pairs are expected to reside, we observed higher quality matches for the $N$-tracked anchors than the $N$-randomly selected anchors (Fig 2A). Therefore, cell tracking is an essential precursor for attaining matches that are both high in quality and spatially coherent. However, when extending this process to all $M$ ($>N$) reference cells, the resulting matches exhibit lower quality due to $M$ exceeding $N$. This trend remains consistent across other cores (S2 Fig). As we know only a fraction of $M$ is physically shared across sections, applying LSA without anchors is not ideal. To illustrate the bulk-level matching without considering spatial information, we employ LSA without using anchors or neighborhood constraints, which we refer to as a variant of MaxFuse without smoothing. MaxFuse's graph smoothing effectively reduces variation within clusters while enhancing differences

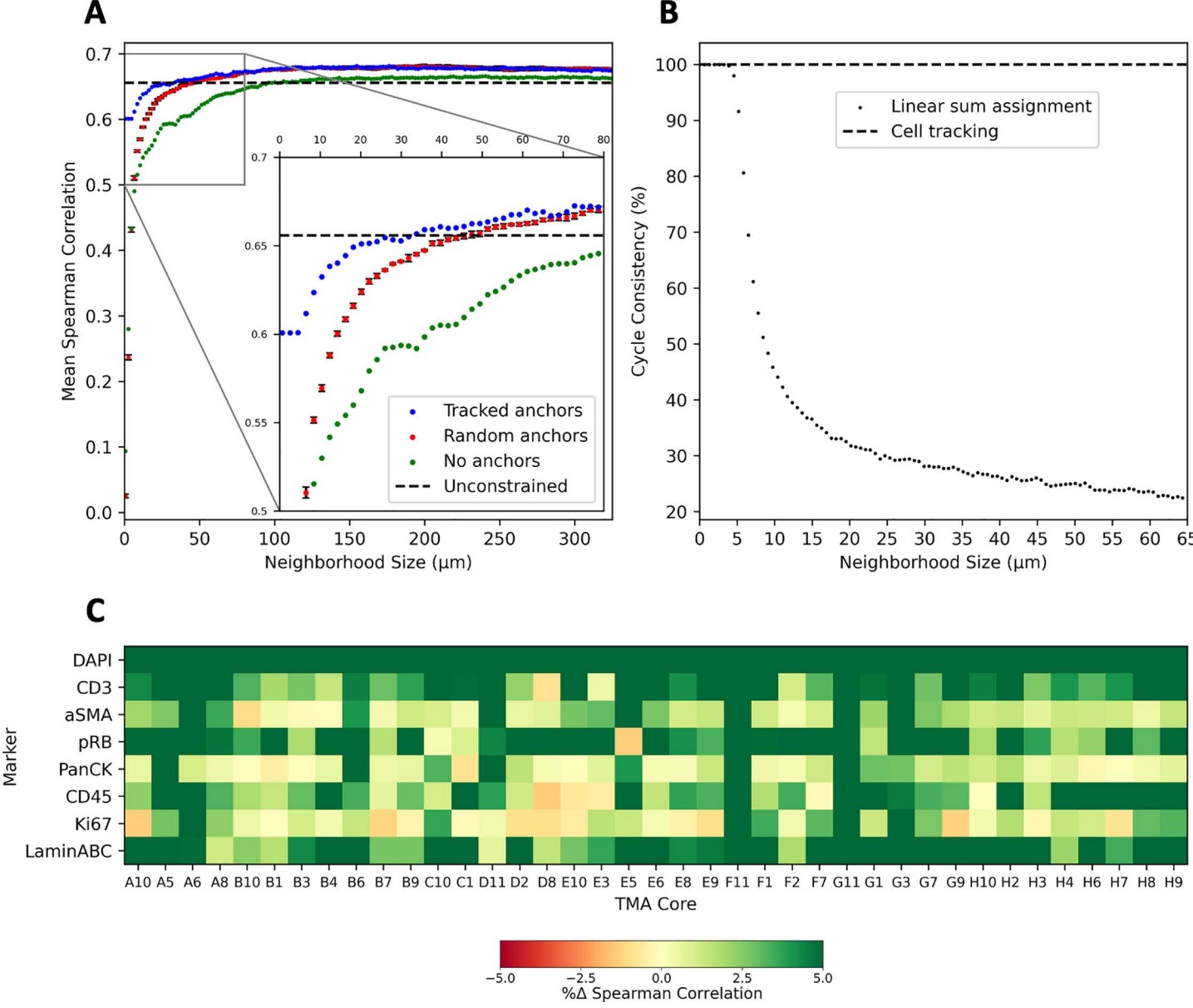

**Fig 2. Cell tracking improves integration quality over linear sum assignment. (A)** Tracked anchors outperform random anchors in reasonably sized neighborhoods. Error bars indicate the standard deviation of five seeds with the constrained case representing bulk-level or LSA without spatial constraints, a variant of MaxFuse. **(B)** Cycle consistency of LSA rapidly diminishes with increasing neighborhood size, whereas cell tracking ensures cycle consistency. **(C)** Heatmap showing the match correlation percent improvement when enriching the tracking algorithm with protein expression information for each TMA core and shared feature, with eight shared markers serving as ground truth for matching and integration. These markers are present in consecutive tissue sections within the dataset.

between them, enabling cluster-level rather than true single-cell matching across modalities. However, this approach is not suitable when true single-cell matching is required. Therefore, we emphasize that our benchmarking is conducted against MaxFuse without smoothing. Interestingly, match quality slightly surpasses this bulk-level correlation at neighborhoods larger than 200µm in the previous unanchored neighborhood expansion scenario (Fig 2A). This occurs because

LSA optimizes linked feature correlation at the individual cell pair level, whereas we assess match quality at the matched population level. While the sum of costs is lower in the bulk-level case, suggesting potentially higher-quality individual matches, it does not translate to higher overall match quality as we measure it.

We also explored the robustness of spatially constrained LSA by measuring its cycle consistency [41], which assesses the algorithm's ability to maintain consistency in assigning matches when swapping the order of the reference and target MTIs. To do this, we matched the original *N*-tracked reference cells with all potential target cells. Then we did the opposite by matching the original *N*-tracked target cells with all potential reference cells. We measured cycle consistency by calculating the fraction of shared matches between both sets relative to *N* (Fig 2B). Our findings revealed a rapid decrease in cycle consistency beyond 5µm neighborhoods, dropping to below 50% at 10µm. This underscores that LSA, widely used as the framework for multimodal integration [36], can yield unstable results when the cost matrix is asymmetric, leading to order-dependent solutions that vary based on the choice of reference and target MTIs and the cell counts in each. If the cost matrix were symmetric, linear sum assignment would yield order-invariant matches. However, in practice, serial sections rarely contain identical numbers of cells, and excluding unmatched cells from one side introduces asymmetry that destabilizes the pairing. In contrast, tracking-based algorithms inherently preserve cycle consistency by design.

## COEXIST

While LSA generates quality matches, empirically fine-tuning neighborhood size is still challenging and subjective. Cell tracking is better suited for this problem because it directly leverages spatial and morphological information and does not require a radius parameter. However, cell tracking does not account for molecular information and is susceptible to erroneously matching cells that overlap but are merely adjacent in the lateral dimension. Therefore, we introduce COEXIST (**Methods**), which combines cell tracking with the molecular feature distance metric used in LSA. By integrating spatial, morphological, and shared feature information, COEXIST provides a robust framework for improving matching accuracy. As we demonstrated the weakness of LSA, we benchmarked COEXIST to the baseline cell tracking algorithm and found it consistently improved the shared feature correlations of the matches, often by more than 5% (Fig 2C). Small decreases in shared feature correlation for specific markers in some cores occurred due to artifacts or tissue loss in the round the markers were stained and imaged.

## Cell type assignment using label propagation and validation

To ensure robust validation of our approach, we use full panel data and synthetically generated two panels to validate COEXIST for refining cell types and label propagation (**Methods**). In this setup, the cell types identified in the full panel serve as the ground truth, enabling us to assess how many anchors are required and evaluate the effectiveness of label propagation (Fig 3A). This experiment was designed to simulate realistic experimental constraints, where only a fraction of cells can be consistently tracked across consecutive slides, and a reduced subset of markers (50% shared markers in the case of the TMA dataset) is available for k-Nearest Neighbors (k-NN) propagation. We tested cell-type label propagation by synthetically generating sub-panels with a random selection of half of the biomarker from the full dataset, simulating the necessity of propagating labels using overlapping markers to untracked cells on consecutive sections.

The experiment was performed on two datasets: (1) a region of interest (ROI) from a healthy tonsil specimen (~70,000 cells) (Fig 3B), and (2) a whole-slide image (WSI) of colorectal cancer (CRC) tissue with adjacent normal tissue (~480,000 cells) (Fig 3C). Results showed that as the proportion of tracked cells increased towards the theoretical maximum of 100% tracking across consecutive sections, label propagation performance, measured by F1 score, improved for both the tonsil ROI and CRC WSI datasets (Fig 3D). For k-NN propagation across 1000 random panels with 60% cell tracking, the mean F1 scores were 0.78 (SD 0.04) and 0.76 (SD 0.03) for the tonsil ROI and CRC WSI datasets, respectively. These scores approached the maximum theoretical mean F1 scores of 0.82 (SD 0.03) and 0.81 (SD 0.02). Importantly, this trend was consistent across varying anchor cell proportions, despite high variations between panels within each anchor

 

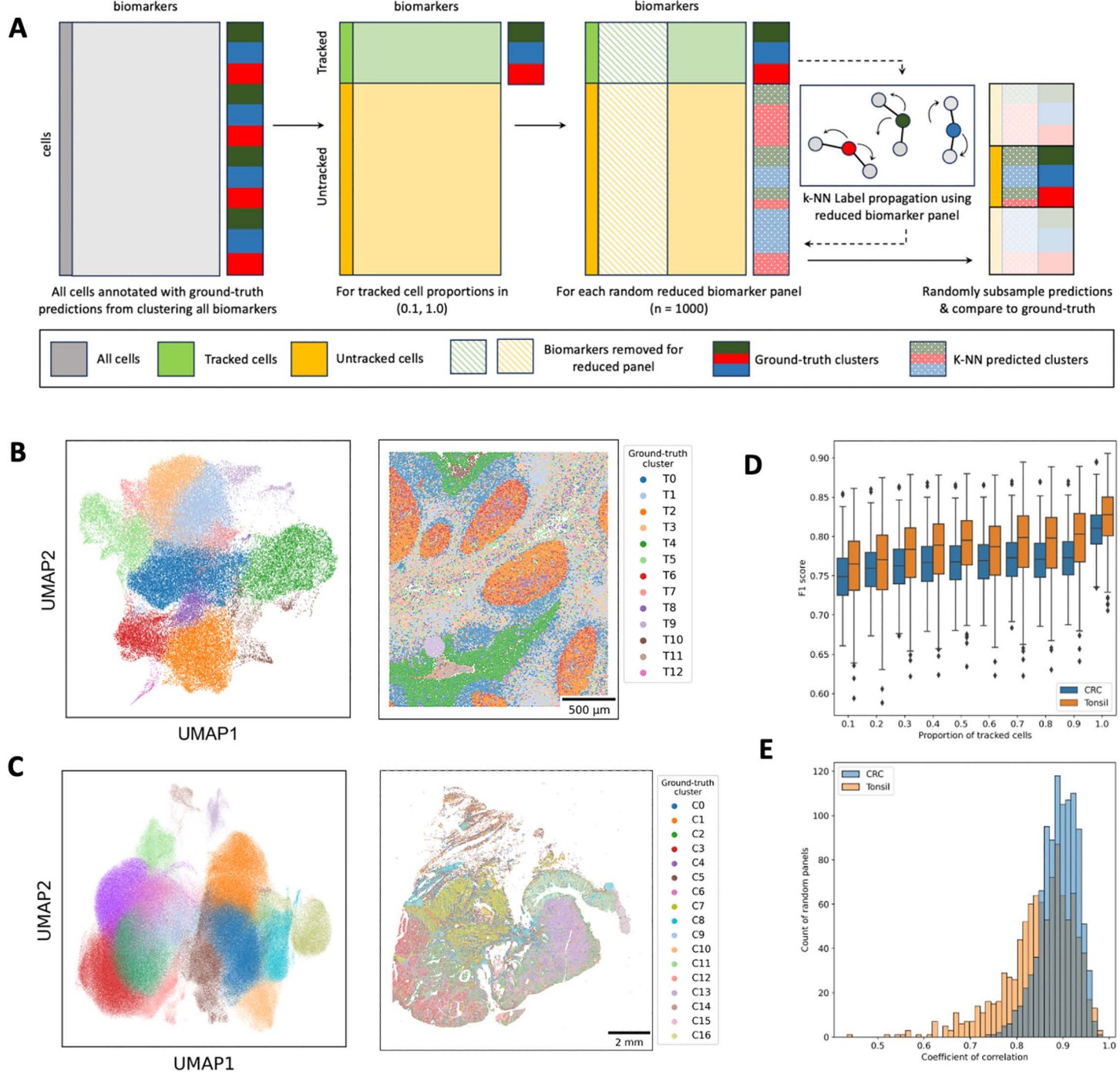

**Fig 3. Label propagation is robust to biomarker panel composition, tracking efficiency, and tissue type. (A)** Overview of single-slide validation experiment. Ground-truth cell type clusters were generated on all cells and all biomarkers for a tissue section dataset. Across a range of proportions, the cells were split into 'tracked' (green) and 'untracked' (yellow) matrices. Within each tracking proportion, the matrices were subset to a random biomarker panel half the size of the original panel. A k-NN classifier was trained on the 'tracked' cells and used to predict which ground-truth clusters an 'untracked' cell belonged to. The predictions were randomly subsampled and compared against the ground-truth labels from the 'untracked' cells. **(B)** Ground-truth Leiden clusters (n = 13) for a healthy tonsil dataset. Cells are annotated by Leiden cluster in a 2D UMAP representation of the full biomarker panel (left) and a spatial scatterplot (right). **(C)** Ground-truth Leiden clusters (n = 17) for a colorectal cancer whole-slide image dataset. Cells are annotated by Leiden cluster in a 2D UMAP representation of the full biomarker panel (left) and a spatial scatterplot (right). **(D)** Boxplot showing F1-score for the

predicted cell type labels at each anchor cell proportion across 1000 random marker panels in the experiment for both the tonsil ROI dataset (orange) and the colorectal cancer WSI dataset (blue) **(E)** Histograms for both the tonsil ROI dataset (orange) and the colorectal cancer WSI dataset (blue) showing the distribution of Pearson correlation coefficients for all 1000 random panels. The correlation coefficient was calculated to test the correlation between anchor cell proportion and F1 score for each panel separately.

cell proportion. Furthermore, both highly performing and poorly performing panels improved as the anchor proportion increased (Fig 3E). These findings highlight that in addition to the proportion of cells tracked across consecutive sections, the specific biomarker composition of the sub-panels plays a critical role in label propagation performance. This underscores the importance of carefully selecting markers to maximize the accuracy and reliability of cell-type assignments across tissue sections.

### COEXIST enhances spatial single-cell profiling

We performed a comprehensive analysis of all BC TMA cores, using combined tumor and immune CyCIF panels with a total of 42 unique biomarkers (S1 Table). We observed several discrepancies in the uni-panel phenotypes compared to the combined panel (Fig 4A). Specifically, cells profiled with the immune panel showed a predominant immune cell population, whereas the tumor panel did not exhibit such dominance. However, the combined panel revealed a more balanced population between these two cases, highlighting COEXIST's ability to refine phenotypes. BC subtype-specific differences contribute to certain batch effects, making an integrative analysis challenging without rigorous normalization [42]. Nevertheless, high-level phenotyping showed evident biases towards the dominant cell population being profiled for a given panel, (i.e., immune or tumor cells), which COEXIST effectively corrects.

We then conducted a thorough phenotypic analysis focusing on triple-negative BC (TNBC) core E6 with combined tumor and immune CyCIF panels comprising 42 biomarkers (S1 Table). We identified 13 distinct cellular phenotypes, unveiling spatial heterogeneity at an unprecedented level (Fig 4B and 4C). However, clustering cells based solely on either the immune or tumor panels failed to capture their full phenotypic diversity (Fig 4B). For instance, the immune panel lacked markers such as CK14, CK17, PCNA, CD31, and Vimentin, which restricted its ability to delineate different epithelial sub-states, vasculature, or cells that may be undergoing Epithelial-to-Mesenchymal transition. Meanwhile, the tumor panel could not differentiate between CD8+ or CD4 + T-cells and lacked markers entirely for B-cells and all myeloid lineage cells including macrophages and monocytes. The combined panel rectified these shortcomings and even revealed a distinct stroma cell population. Shared markers across the panels clustered closely (Fig 4C) with moderate Spearman correlations (S3 Fig).

To further investigate cell identity switching, we examined a population of 138 cells that were classified as αSMA+ stromal cells in the tumor panel but reclassified as macrophages when the immune panel was incorporated (Fig 4D). For this population of cells, lineage marker intensities were ranked against the remaining cells in the core using each panel. In the tumor panel, αSMA and Vimentin emerged as the highest-ranked markers. Meanwhile, in the immune panel, markers such as CD14, MHC II, CD11b, αSMA, CD45, CD163, and CD68 were among the most prominently ranked, driving the identification of the macrophage population. On the combined panel (depicted in purple for markers originating from the tumor panel and green for those from the immune panel in Fig 4D), CD14 and MHCII were notably more differentially abundant compared to αSMA. These differences likely stem from the absence of myeloid-specific lineage markers in the tumor panel. Without these lineage-specific markers, fluorescent spillover from nearby cells or high autofluorescence in other channels makes the misclassification of cell types more likely (S4 Fig). The integration of both panels effectively resolved this discrepancy. Apart from these markers, profiles across the cells were generally similar.

Next, we propagated the rich phenotype labels to the unshared cells in each section using k-NN (**Methods**), showing increases in cell type diversity for each panel (Fig 4E). The immune panel inherits increased cell type resolution within the epithelial and stromal compartments, while the tumor panel shows a large expansion in immune cell type resolution.

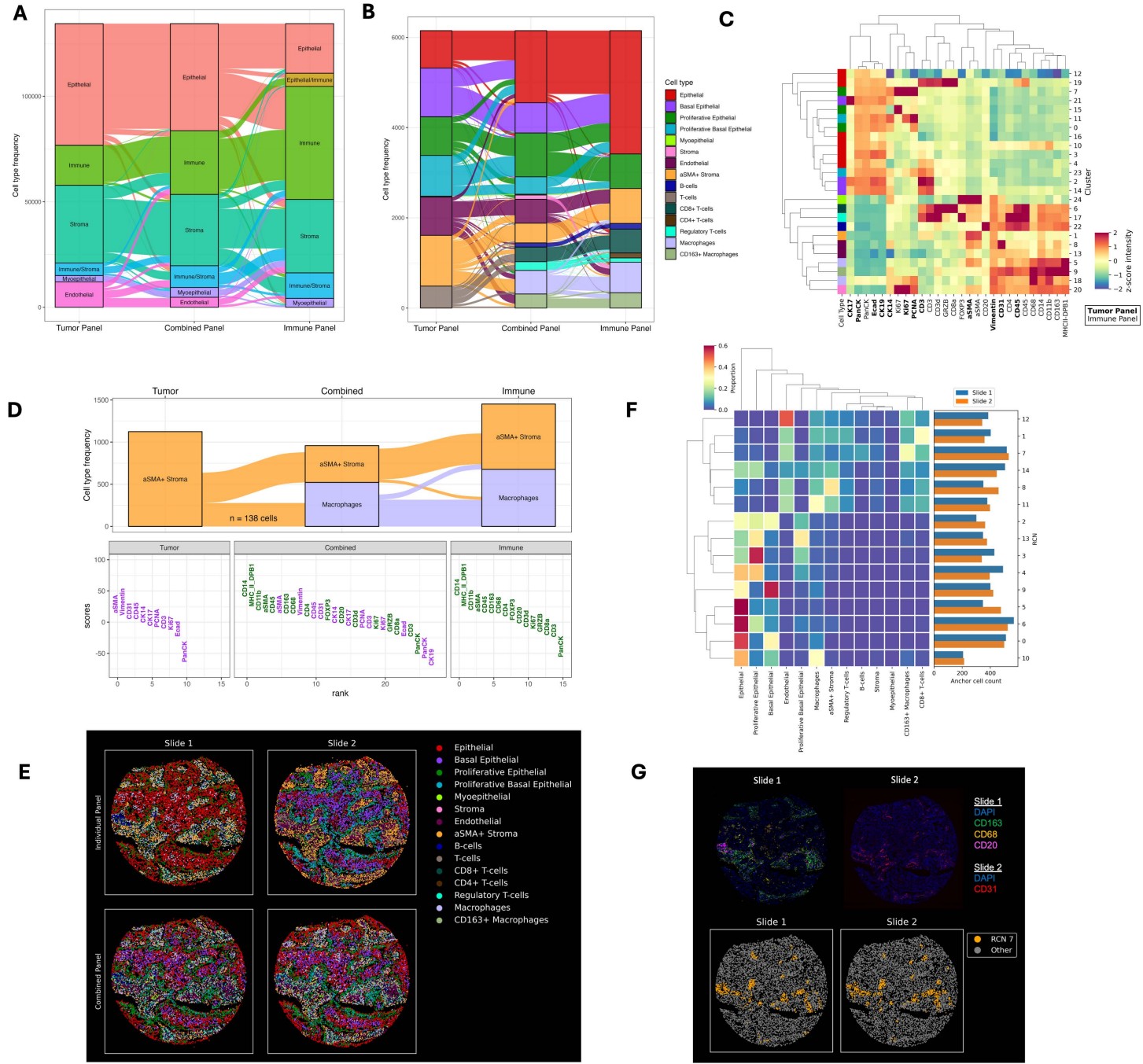

**Fig 4. COEXIST enables rich spatial profiling of BC TMA cores. (A)** Alluvial plot of high-level cell phenotype assignments of the matched cells generated by COEXIST, using either the tumor panel, immune panel, or combined panel for 39 BC subtype TMA cores. **(B)** Alluvial plot of refined cell types for E6 phenotyped with either the immune panel, tumor panel, or combined panel. **(C)** Heatmap of Leiden clusters used to define refined cell types of the matched cells in triple negative BC core E6 based on the combined panel. **(D)** Alluvial plot (top) and biomarker rank plot (bottom) highlighting a subset of tracked cells (n = 138) that switched cell type identities between the panels. For the biomarker rank plot, lineage markers were ranked for these 138 cells against the rest of the cells in the core. Purple represents biomarkers derived from the tumor panel, while green represents immune panel biomarkers. **(E)** Visual comparison of the cell phenotypes mapped to their spatial locations in the TNBC core consecutive sections (columns) before (top row) and after (bottom row) integrating them with COEXIST. **(F)** Recurrent neighborhood analysis heatmap showing cell composition per neighborhood and each neighborhood's frequency in both sections. **(G)** Case study of RCN 7 from **(F)** showing image channels corresponding to RCN 7's markers in both slides (top row) and the presence of RCN 7 after applying COEXIST (bottom row).

The resulting spatial distribution of cell types in the two sections appeared very similar after label propagation (Fig 4E). To quantify this similarity, we determined the cell type composition of all cells utilizing the k-NN propagation. The propagated cell composition remained consistent across panels when compared to their original counterparts (S5 Fig). We also documented recurrent cellular neighborhoods around cells shared by the two panels. Our recurrent neighborhood (RCN) analysis showed consecutive sections varied in the abundance of certain neighborhoods, such as RCN 3, 4, 5, and 8 (Fig 4F). This highlights the weakness in assuming spatial homogeneity across the tissue block. Other neighborhoods are similarly abundant across the two panels. One such neighborhood, RCN 7, is composed primarily of CD163 + Macrophages, CD31 + Endothelial cells, and CD20 + B-cells (Fig 4G). These lineage markers are spread across the two panels, and therefore the co-localization of these cell types would have been missed without the combination of the two panels. This RCN may suggest that vascular regions play an important role in immune cell infiltration. Further, this example highlights the utility of panel combination to uncover potentially biologically meaningful functional tissue units.

**COEXIST elevates cross-MTI platform validation**

Validating across MTI platforms is critical for ensuring scientific reproducibility in spatial biology. We demonstrated COEXIST's effectiveness in comparing MTI platforms using serial sections from normal jejunum core B4 in the TNP-TMA BC dataset processed with CyCIF and multiplex immunohistochemistry (mIHC) [17]. We compared the section bulk-level population compositions and found a strong Pearson correlation ($r = 0.8$) between the platforms (Fig 5A). This result is expected and has been demonstrated by other groups [4]. We performed the population comparison exclusively on the subset of shared cells identified by COEXIST, observing a similar correlation ($r = 0.85$) in compositions (Fig 5B). However, recognizing the limitation of bulk-level comparisons, we assessed the platforms at the single-cell level and found a moderate correlation, with Spearman correlations ranging from 0.26 (CD11B) to 0.81 (PanCK) (Fig 5C). The lower Spearman correlations observed at the single-cell level compared to the population level can be attributed to the limited presence of positive populations for certain markers (i.e., rare cell types). Additionally, we noticed the impact of the nonlinearity introduced by the IHC enzyme reaction, which affects signal conversion to expression levels.

## Discussion

While staining a single tissue with both panels would be ideal, it is often impractical due to tissue loss and signal degradation caused by multiple rounds of CyCIF. To address this, our method introduces an innovative cell tracking and propagation framework that integrates complementary antibody panels across sections. This approach expands the phenotypic space without requiring all markers to be present in a single section. Building on this, we recognize that prevalent multi-omic integration methods often do not pair the same cells across modalities; instead, they rely on matching cells with similar molecular states using limited shared information. However, our study highlights that characterizing molecular states based solely on shared marker information for matching is not sufficient to capture the true spatial relationships between cells, and further, spatial information is crucial for precise single-cell matching.

To combat this challenge, we developed COEXIST and demonstrated its utility for precise single-cell matching across serial sections. Using COEXIST, we resolved differences between two orthogonal MTI panels with limited marker overlap and establish a set of refined cellular phenotypes on breast cancer tissues from a Human Tumor Atlas Trans-Network Project (TNP) Tissue Microarray. Further, we successfully establish detailed cell phenotypes for one triple negative breast cancer core. While individual panels can cause misclassification of some cell populations across serial sections, due to the absence of key lineage markers, COEXIST rectified these discrepancies. Our neighborhood analysis revealed unique and potentially biologically significant spatial structures, such as a neighborhood enriched for vascular cells and a diverse complement of immune cells, that could be captured only by the combined panel. Additionally, COEXIST mitigated the problem of overcalling specific lineages that is common in panels designed for specific cell types like immune cells,

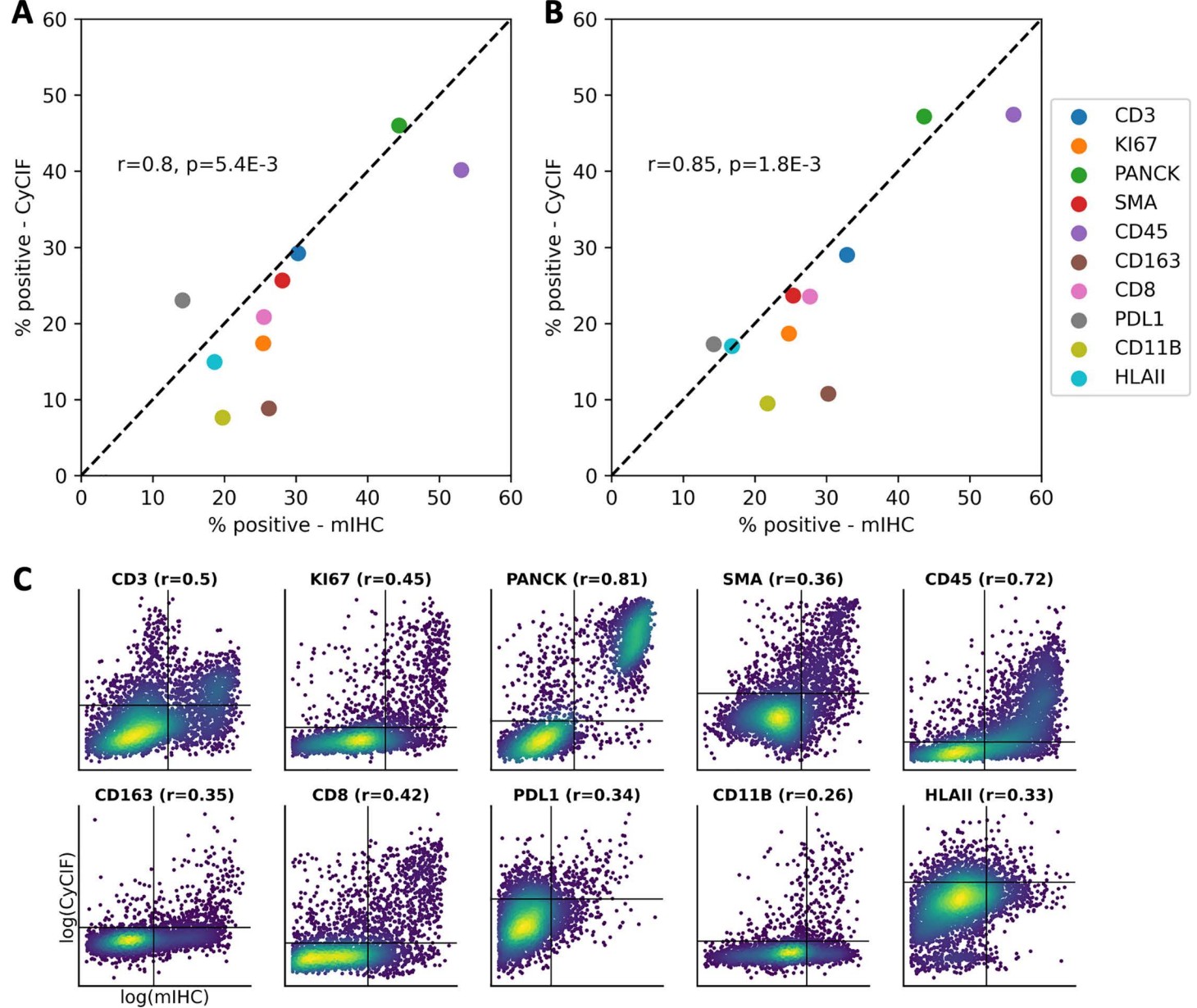

**Fig 5. COEXIST provides MTI platform comparisons at the single-cell level. (A)** Population-level scatter plot and Pearson's *r* of serial TMA sections with (**A**) all cells and (**B**) shared cells identified with COEXIST. **(C)** Single-cell comparison of mIHC and CyCIF platforms with 10 shared markers, shown with Spearman correlation.

fibroblasts, macrophages, and malignant epithelial cells, thereby contributing to more accurate cellular phenotyping across diverse tissues.

We show COEXIST has *O(n)* time complexity with respect to the number of input cells, making it highly scalable to large MTIs (S6 Fig). However, the preprocessing step of computing the pairwise correlation distances is $O(n^2)$, which can be improved by only computing local distances rather than all possible distances. It is important to note that employing COEXIST necessitates the use of two consecutive slides, which should be considered in the context

PLOS Computational Biology

of experimental design and cost management. Despite the cost challenge, applying COEXIST two consecutive slides allows for the joint analysis of about 60% of the nuclei that are intact across two 5µm sections. COEXIST's key assumption is that the shared cell population is representative of the broader tissue population. However, this assumption may not hold in cases where the tissue sections are thicker than the average cell diameter, potentially leading to a disproportionate representation of certain cell populations. For example, the immune cells, which tend to be significantly smaller than malignant cells, could be especially affected depending on how the section is cut in a 2.5-dimension. In this study, we focused on two consecutive sections, but this approach can be expanded to more sections while keeping the reduced sharing of cells in mind. Nonetheless, we analyzed three consecutive sections simultaneously by treating the middle section as our reference and then using pairs of two consecutive sections with COEXIST. Interestingly, the matched cells generated by COEXIST show only moderate shared feature correlations, which is unexpected considering the direct linkage of protein marker expression through tissue processing and antibodies. This contrasts with multi-omics, where the linkage between features is weaker and studies exploring differences in paired multi-omics measurements often reveal discrepancies [43,44]. Several factors contribute to these observed discrepancies; First, many shared markers identify rare cell populations, leading to most cells being "negative" (Fig 5C). Second, compartment-level differences may exist within the same cell, as the cells are being measured in different regions across consecutive sections. Third, technical batch effects are expected between MTI platforms, for instance, mIHC uses enzymatic labels for molecule tagging, whereas CyCIF employs immunofluorescent labels [45]. Comparatively, CyCIF-CyCIF integration exhibits stronger linear relationships in the expression levels of shared markers (S7 Fig). These factors should be considered in future multi-omics integration studies.

In addition to requiring two consecutive sections, COEXIST assumes high-quality image registration and that spatial proximity and shared marker expression serve as reliable anchors for matching. To test COEXIST's robustness to registration errors, we rotated and translated one section and re-applied COEXIST, measuring how many original cell pairs were recovered. We found rotations of <0.3 radians and translations of <4 pixels resulted in 80% or more recovery (S8 Fig). Users should ensure quality registration through visual inspection and cross-referencing the matching percent with the Monte Carlo simulation framework. We do not impose strict assumptions about the distribution of cell features or spatial organization, acknowledging that these characteristics can vary widely across tissue types. Despite this flexibility, we recognize that this assumption may introduce biases, particularly in complex tissues with highly heterogeneous cell distributions. We also acknowledge the limitation of using shared feature correlation as a metric for match quality. However, this approach aligns with other publications [12,35] for assessing match quality. Publicly available datasets suitable for validating COEXIST across diverse tissue types remain sparse. To address this, we experimented using two single-slide datasets: one from a healthy tonsil specimen and another from a colorectal cancer (CRC) tissue sample. These datasets were used to validate the propagation of cell type labels under varying degrees of cell tracking efficiency and across random biomarker panels. The two tissue types represent very different spatial organizations — tonsil tissue is highly spatially stratified by cell type, while CRC tissue is more spatially mixed. Notably, both tissue types showed improved label propagation performance as cell tracking efficiency increased. To establish an upper bound for label propagation performance, we simulated perfect cell tracking by using 100% of cells for k-NN training. In this scenario, label propagation errors could only be attributed to missing key biomarkers in reduced panels or shared markers across consecutive sections. This finding highlights the importance of marker selection, especially for future multimodal applications where feature correspondence across modalities may be limited.

While existing 3D segmentation tools such as Mesmer and Cellpose have been successfully applied to thin sections (2–3µm), their applicability is more limited in thicker sections (e.g., 5 µm) or suboptimally prepared tissue where nuclear preservation across sections may be incomplete. COEXIST addresses these challenges by balancing biomarker profiles with spatial proximity to assign matches, improving flexibility in heterogeneous tissues. These features make COEXIST particularly well-suited for retrospective or opportunistic 3D reconstruction where sample preparation was not optimized for volumetric analysis. Moreover, COEXIST can complement existing 3D segmentation approaches by refining tracking

results or propagating cell type annotations based on molecular similarity, especially in regions where segmentation continuity is disrupted. Looking forward, 3D MTIs with high lateral resolution can be used for ground truth validation of COEXIST when they become available.

Alternative computational integration methods employ strategies such as dimension reduction or graph smoothing to reduce complexity in high-dimensional data. However, MTI panels are relatively compact compared to broader single-cell assays and are often designed to measure orthogonal features, such as the tumor and immune panels in the TNP-TMA BC dataset. Therefore, those strategies are not required for our applications in this paper. For tasks requiring high-dimensional multimodal integration, COEXIST is highly compatible with joint embedding approaches [30,36], making it adaptable to a wide range of spatially resolved datasets. COEXIST is designed to allow researchers to fully leverage MTI technologies like CyCIF and mIHC and improve MTI assay development. While MaxFuse demonstrated that cell matching is possible between modalities with weakly linked feature spaces, we have shown that these matches are prone to be spatially incoherent. Given its flexible framework and ability to be applied to any spatially resolved assay, future use cases for COEXIST could include spatially aware integration across modalities with weakly linked features, such as spatial proteomics and spatial transcriptomics.

## Methods

### Preprocessing

The TMA dearray [46], illumination correction, tile alignment, and stitching were conducted within the Galaxy-MCMICRO Environment [5]. The cells and nuclei were segmented using common nuclear (DAPI) and membrane markers (Pan-cytokeratin, CD45) with Mesmer [47]. Corresponding cells and nuclei were assigned the same label, and cells without a corresponding nucleus were removed. Consecutive nuclear masks were registered using MATLAB's imregister function [48]. The nuclei pixel coordinates, centroid, and marker mean intensities were extracted with scikit-image [49].

### Quality control

Quality control procedures were implemented to ensure the integrity of the TMA cores. Out of the original 88 cores (3 consecutive tissue sections per core), 49 were removed due to various issues encountered during imaging and processing, including fluorescent artifacts in the images, tissue loss, and registration failures. Notably, the majority of registration failure cases of samples were not consecutive sections, indicating a significant difference in tissue architecture. Since our approach depends on consecutive tissue, we use quality-controlled cores to showcase COEXIST.

### Monte Carlo simulation

To empirically estimate nuclei sharing across consecutive tissue sections, we conducted a Monte Carlo simulation (Algorithm 1). Our analysis is based on three main assumptions about the system:

(1) The nuclei are spatially uniformly distributed. (2) The nuclei are spherical with normally distributed diameters. (3) Nuclei are present in a section if their boundary overlaps the section boundary.

```
Algorithm 1: Monte Carlo simulation to estimate nuclei sharing across consecutive sections.

Input:
    I: number of repetitions
    M: nuclear diameter mean (μm)
    S: nuclear diameter standard deviation (μm)
    W: section thickness (μm)
    B: minimum and maximum tissue thickness (μm)
Initialize:
    n_shared = 0: number of shared nuclei in sections 1 and 2
```

```
n_unshared = 0: number of unshared nuclei present in section 1
for i from 1 to I do
    draw nucleus radius r from normal distribution parameterized by M and S
    draw nucleus coordinate x from uniform distribution bounded by B
    get nucleus bounds n_min, n_max = x ± r
    if n_min < W < n_max then
        n_shared += 1
    else if 0 < n_max < W then
        n_unshared += 1
    else
        continue
    end if
end for
f_shared = n_shared/ (n_shared + n_unshared)
return f_shared
```

## Cell tracking

We perform single-cell tracking using majority voting and cycle consistency. Our method is a two-step process: first, we perform a majority vote from the reference slide to the target slide. Then, we ensure cycle consistency by considering the target slide back to the reference, guaranteeing uniqueness in the matching procedure for tracking.

Formally, given sets P and Q, representing nuclei present in serial tissue section images $I_P$ and $I_Q$, we aim to find $P \cap Q$ defined as subsets $A \subseteq P$ and $B \subseteq Q$ such that elements $a_i \in A$ and $b_i \in B$ represent the same nucleus present in both sections, where the number of cells n(A) = n(B). We begin with nuclear segmentation label masks $M_P \in Z^{m1 \times n1}$ and $M_Q \in Z^{m1 \times n1}$. To match a nucleus $a_i \in A$ to nucleus $b_i \in B$, the tracking algorithm first identifies nuclear region $r_{ai} \in M_P$ representing the pixel coordinates of $a_i$, ensuring spatially coherent match candidates. Next, nuclei b with regions $r_b \cap r_{ai}$ are considered for matching and nucleus $b_i \in b$ is assigned as the nucleus with the maximum region overlap between $a_i$ and candidates b. To ensure cycle consistency, the reverse process is completed (assigning $b_i$ to candidates a). If the subsequent assignment is not $a_i$, the nuclei are considered unmatched. In the case where multiple candidates in b have equivalent region overlaps of $a_i$, the candidate with the minimum centroid distance to $a_i$ is chosen.

## Linear sum assignment (LSA)

LSA offers an alternative framing of the problem by finding a linear matching between sets P and Q such that the sum of the match costs is minimized. We begin with cell-feature arrays $F_P \in Z^{m2 \times n2}$ and $F_Q \in Z^{m3 \times n2}$, where features of $F_P$ and $F_Q$ are the shared molecular features. $F_P$ and $F_Q$ are standardized to a mean of 0 and standard deviation of 1 to remove intensity range variations. The pairwise Spearman correlation distance, 1 – Spearman's $\rho$, between nuclear pairs $D \in R^{m2 \times m3}$ is used as the cost matrix. While Spearman's correlation is a non-parametric similarity measure rather than a strict distance metric, we treat 1 − Spearman's $\rho$ as a distance-like measure. This allows positive correlations to correspond to low costs and negative correlations to high costs, aligning with the optimization goal. Spearman's $\rho$ was chosen over Pearson's $\rho$ because it is robust to outliers and measures the strength of monotonic relationships between marker expression profiles. Additionally, in the context of multiplex tissue imaging (MTI), mean intensity feature correlation is commonly used, which motivated the choice of Spearman correlation over alternatives such as cosine distance. However, we acknowledge the potential benefits of cosine distance and may explore it in future work.

A limitation of this approach is that it does not guarantee matches $a_i$, $b_i \in A$, B are spatially coherent. To address this, we incorporate spatial information by modifying D before applying LSA by assigning infinite cost to nuclear pairs that have intercellular distances greater than an empirically determined radius using the pairwise X-Y projection of the Euclidean centroid distance $E \in R^{m2 \times m3}$. The SciPy implementation [49] of the modified Jonker-Volgenant algorithm [39] was used for this purpose, which extends the JV algorithm to the rectangular assignment problem where the bipartite graph has an

unequal number of nodes in each class. This modification is necessary because there are often unequal numbers of cells across modalities/tissue sections. This algorithm is compatible with any cost matrix representing the "cost" of assigning nodes from different classes to each other. Costs can be positive, negative, or zero, and the goal of the algorithm is to find a 1:1 matching such that the sum of the overall costs is minimized.

## COEXIST

We propose COEXIST for integrating serial MTI slides of reasonable thickness. This algorithm combines spatial and biomarker expression information to match single cells present in both slides. Across a set of spatially overlapping cells, COEXIST assigns matching cells by minimizing the 1-Spearman correlation coefficient between their molecular profiles (Algorithm 2). Therefore, spatially coherent candidate cells are considered, but matches are assigned by shared molecular profiles. In scenarios where there exists no molecular linkage $F_P$ and $F_Q$, cell tracking can be used.

```
Algorithm 2: COEXIST to match shared nuclei across consecutive MTIs.

Input:
    M1: nuclear segmentation for serial MTI 1
    M2: nuclear segmentation for serial MTI 2
    D: cost matrix of pairwise (1 − Spearman's ρ)
Initialize:
    B=set: empty bijective set for cell pairs in M1 and M2
    for each cell i in M1 do
        get pixel coordinates of cell i
        get unique pixels (cells) with cell i's pixel coordinates in M2
        for cell j in unique pixels do
            get cost of pair D[i, j]
        end for
        assign cell i to cell j with minimum cost
        if cell i assigned to cell j then
            get pixel coordinates of cell j
            get unique pixels (cells) with cell j's pixel coordinates in M1
            for cell k in unique pixels do
                get cost of pair D[k, j]
            end for
            assign cell j to cell k with minimum cost
            if cell k == cell i:
                add pair (i, j) to B
            else:
                continue
            end if
        end if
    end for
    return B
```

## Clustering and label propagation

After identifying the shared subpopulation, we cluster the cells with GPU-accelerated Phenograph [41] and propagate the labels to the remaining cells in each slide. Clustering parameters *n_neighbors* were set to 60, 70, and 100 for the tumor panel, immune panel, and shared subpopulation, respectively, and *min_size* was set to 100 for all. For all clustering performed in our analyses, we explored a range of different clustering granularities in order to find the resolution that generated clusters for which we were able to reliably annotate as biologically meaningful cell types. To achieve label

propagation to the unshared subpopulation of each section, we fit a GPU-accelerated k-NN classifier from the RAPIDS library [50] on the panel-specific markers of the shared subpopulation along with their phenotype labels. For each cell of the unshared subpopulation, we use the panel-specific k-NN classifier to query and majority-assign the superplexed labels of its 10 nearest neighbors in the shared subpopulation, determined using the default Euclidean distance.

### Validation of cell-type label propagation

For both the tonsil ROI and CRC WSI datasets, quantified cellular feature tables were downloaded from publicly available repositories. Each dataset included 24 relevant protein markers after excluding DAPI and autofluorescence channels. Marker intensities were log1p transformed, and a standard scaler was applied using Scanpy. Subsequently, Scanpy was used to generate a neighborhood graph, compute a UMAP representation, and perform Leiden clustering on each dataset separately. The resolutions used for Leiden clustering were determined empirically, with a value of 0.5 identified as optimal for capturing biologically meaningful cell types in both datasets. This resulted in 13 cell types for the tonsil ROI and 17 cell types for CRC WSI datasets. The resulting Leiden cluster labels, prefixed with "T" for the Tonsil ROI dataset and "C" for the CRC WSI dataset, were used as ground-truth labels for the following experiments.

To simulate tracking variability, we generated 1,000 random panels, each containing 12 markers selected from the original set of 24 markers. The cell feature tables were then divided into 'tracked' and 'untracked' matrices across a range of proportions, from 10% to 100%. This range represents varying levels of tracking efficiency, with 10% corresponding to poor tracking and 100% representing perfect tracking. Based on our data, a realistic expectation for tracking efficiency is approximately 60% of cells shared between consecutive tissue sections.

The 50% reduction in panel size reflects practical limitations in shared marker availability, simulating the realistic training data size for k-NN label propagation to untracked cells. This setup mirrors scenarios encountered in TMA data, where 48 markers are used for cell phenotyping and 24 markers are available for k-NN per panel. Consequently, targeting 50% marker usage for our simulations reflects real-world constraints on shared biomarker panels.

For each anchor cell proportion and random panel, a k-NN classifier was trained on the 'tracked' matrix using only the intensities of the 12 random markers and the ground-truth Leiden cluster annotations as training labels (Fig 3A). To ensure consistency across anchor cell proportions and emphasize local cell relationships, the number of neighbors ($n$_neighbors) was set to 10 for all cases. The trained classifier was then used to predict Leiden cluster labels for cells in the 'untracked' matrix. To evaluate performance, a subset of predicted labels (5,000 cells for the tonsil ROI and 40,000 cells for the CRC WSI dataset) was sampled and compared to the ground-truth annotations to compute an F1 score. F1 scores were calculated as a weighted average across Leiden cluster labels for each random panel. For the 100% tracked scenarios, a random sample of the tracked cells was used for both prediction and scoring to establish a theoretical maximum performance benchmark.

### Cell phenotyping

The Phenograph-derived clusters were used to assign broad cell types across all breast cancer and normal breast tissue cores that passed quality control; however, this combined clustering lacked sufficient resolution to classify cell types beyond high-level lineages (Epithelial, Immune, Stroma, etc.). To conduct more detailed analyses, one triple-negative breast cancer core from the TMA (E6) was selected and subjected to independent cell phenotyping. For individual core phenotyping, Scanpy [51] was used to cluster cells based on a subset of lineage-specific markers from the tumor, immune, and combined panels. A neighborhood graph was constructed using default parameters (*scanpy.pp.neighbors*), and clusters were assigned using Leiden clustering (*sc.tl.leiden*), with the *resolution* parameter set to 2. Leiden clusters were annotated and binned into cell types based on the mean intensities of lineage markers for each of the three panels. Propagation of core-specific labels from the combined panel to the unshared population of each panel was conducted in

the same manner as described in the previous section. To characterize a subpopulation of cells that switched identities between panels, lineage markers were ranked (*scanpy.tl.rank_genes_groups*) for the subpopulation against the remaining cells in the core for each panel.

### Recurrent neighborhood analysis

For core E6, a Recurrent Cellular Neighborhood (RCN) analysis [52] was conducted to examine the spatial arrangement of cell types between the two panels after label propagation. For each panel, anchor cells for the neighborhood analysis were considered as those that were shared across the panels. For each panel independently, a Scikit-learn [53] KDTree was used to calculate a counts matrix of neighboring cell types within 30µm of anchor cells. The neighborhood counts were converted to proportions, and these relative neighborhood matrices from the two panels were concatenated together. Neighborhoods were clustered using K-means clustering into 15 unique RCNs. Values in the range of 5–50 were used to construct an elbow plot to determine the final number of clusters for the k-means clustering.

### Data visualization

The *ggplot2* [54] and *ggAlluvial* [55] packages were used to generate stacked bar plots and alluvial diagrams in R [56]. All other plots were generated using the seaborn [57] and matplotlib [58] packages in Python. Visualizations of the actual multiplex tissue images were captured using Vitessce [59,60] within the Galaxy-MCMICRO Environment [5].

### Statistics and reproducibility

We utilize both Pearson and Spearman correlation analyses in our study to explore variable relationships. Pearson correlation assesses linear connections among continuous variables, while Spearman correlation evaluates monotonic relationships.

To guarantee result reproducibility, we provide all the necessary code snippets and scripts used in our analysis. This includes data processing steps, statistical analyses, and visualization methods.

### Supporting information

**S1 Fig. Out-of-the-box linear sum assignment does not recover spatial information. (A)** y-coordinates of linear sum assignment matches connected by lines across Slide 1 and Slide 2 of core B3 shown on the y-z projection of serial slides. **(B)** y-coordinates of cell tracking matches connected by lines across Slide 1 and Slide 2 of core B3 shown on the y-z projection of serial slides.
(TIFF)

**S2 Fig. Tracked anchors consistently outperform random anchors.** Shared marker correlation of cell pairs versus search radius for linear sum assignment matching based on tracked *N* anchors, random *N* anchors, and no anchors for cores H7, F1, H4, G6, E5, and A10.
(TIFF)

**S3 Fig. COEXIST's performance across the TNP-TMA dataset.** Mean Spearman correlation of 39 TMA cores for 8 shared markers.
(TIFF)

**S4 Fig. Missing lineage markers in core E06 result in cell identity switching between panels.** A subset of tracked cells switched cell type identities between the panels (Fig 4D). To investigate this further, we focused on one cell type identity transition. Among the 138 tracked cells, those classified as αSMA+ stroma cells in the tumor panel were

reclassified as macrophages on the Immune and Combined panels. The images highlight how these misclassifications can occur. In the immune panel image (right), CD68 (green; canonical macrophage/monocyte biomarker) is clearly abundant and closely co-located with αSMA+ (magenta). As a result, these cells are identified as macrophages in the Immune panel due to the presence of CD68. However, the tumor panel (left) lacks the CD68 marker, leading to their classification as αSMA+ stromal cells. This misclassification may arise from signal spillover from overlapping or nearby stromal cells, or from high autofluorescence in the αSMA channel. By combining the panel, this discrepancy was resolved.
(TIFF)

**S5 Fig. Cell type composition comparison and their transition across panels.** Alluvial plot of propagated refined cell types for E6 phenotype with either the immune panel, tumor panel, or combined panel.
(TIFF)

**S6 Fig. COEXIST's computational demands.** Runtime of the COEXIST algorithm with respect to the average number of input cells across consecutive MTIs using core B4 in the TNP-TMA dataset.
(TIFF)

**S7 Fig. COEXIST's performance across adjacent sections of the same MTI technology.** Single-cell scatter plots of marker intensity across shared markers in consecutive CyCIF MTIs for core B4.
(TIFF)

**S8 Fig. COEXIST is robust against small registration errors.** Match recovery after applying rotation and translation transforms to section 2 for core B4.
(TIFF)

**S1 Table. TNP-TMA BC biomarker panels.**
(XLSX)

## Acknowledgments

The resources of the Exacloud high-performance computing environment developed jointly by OHSU and Intel and the technical support of the OHSU Advanced Computing Center are gratefully acknowledged.

## Author contributions

**Conceptualization:** Young Hwan Chang.

**Data curation:** Robert T. Heussner, Cameron F. Watson, Kunlun Wang, Eric M. Cramer.

**Formal analysis:** Robert T. Heussner, Cameron F. Watson, Christopher Z. Eddy, Allison L. Creason.

**Funding acquisition:** Gordon B. Mills, Young Hwan Chang.

**Methodology:** Robert T. Heussner, Cameron F. Watson.

**Project administration:** Young Hwan Chang.

**Resources:** Young Hwan Chang.

**Software:** Robert T. Heussner, Cameron F. Watson.

**Supervision:** Young Hwan Chang.

**Visualization:** Robert T. Heussner.

**Writing – original draft:** Robert T. Heussner, Cameron F. Watson, Allison L. Creason, Gordon B. Mills, Young Hwan Chang.

**Writing – review & editing:** Robert T. Heussner, Cameron F. Watson, Christopher Z. Eddy, Kunlun Wang, Eric M. Cramer, Allison L. Creason, Gordon B. Mills, Young Hwan Chang.

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
