## [Decision Letter · Decision Letter 0]

29 May 2025

COEXIST: Coordinated single-cell integration of serial multiplexed tissue images

PLOS Computational Biology

Dear Dr. Chang,

Thank you for submitting your manuscript to PLOS Computational Biology. After careful consideration, we feel that it has merit but does not fully meet PLOS Computational Biology's publication criteria as it currently stands. Therefore, we invite you to submit a revised version of the manuscript that addresses the points raised during the review process.

Please submit your revised manuscript within 60 days Jul 29 2025 11:59PM. If you will need more time than this to complete your revisions, please reply to this message or contact the journal office at ploscompbiol@plos.org. Please include the following items when submitting your revised manuscript:

We look forward to receiving your revised manuscript.

Kind regards,

Marc Birtwistle

Section Editor

PLOS Computational Biology

**Journal Requirements:**

1) Please provide an Author Summary. This should appear in your manuscript between the Abstract (if applicable) and the Introduction, and should be 150-200 words long. The aim should be to make your findings accessible to a wide audience that includes both scientists and non-scientists. Sample summaries can be found on our website under Submission Guidelines:

3) We notice that your supplementary Figures, and Tables are included in the manuscript file. Please remove them and upload them with the file type 'Supporting Information'. Please ensure that each Supporting Information file has a legend listed in the manuscript after the references list.

4) Please amend your detailed Financial Disclosure statement. This is published with the article. It must therefore be completed in full sentences and contain the exact wording you wish to be published. Please ensure that the funders and grant numbers match between the Financial Disclosure field and the Funding Information tab in your submission form. Note that the funders must be provided in the same order in both places as well.

**Reviewers' comments:**

Reviewer's Responses to Questions

**Comments to the Authors:**

Reviewer #1: In this work, Heussner, Watson et al introduce COEXIST, a method for integration of cellular data across serially sectioned slides. This method shows great potential for improving multiplex integration for the many serial sectioning datasets emerging within the tissue atlas community. Below I provide detailed comments:

In the introduction, the authors write “to our knowledge, no algorithms or computational frameworks explicitly designed to pair single cells across consecutive tissue sections currently exist.” I advise avoiding statements claiming to be the first to do something, as this is hard to prove. As one counter example, I identified ref [1], a manuscript published in 1988 presenting a software solution for reconstruction of nuclei across serial electron microscopy images. Likely more recent algorithms exist to do similar intra-slide cellular comparison.

The authors determined that their dataset possessed a mean nuclear diameter of 7.8 micron, then used this diameter to simulate spherical nuclei for the cell tracking algorithm implementation. What are the consequences of this averaging for very small nuclei (such as lymphocytes) compared to very large nuclei (such as breast cancer cells). The error for these different cell types is likely very different. Lymphocytes may at times only exist in a single tissue section, while normal epithelial or cancer cells will nearly always exist in 2 – 3 sections. How does the accuracy of COEXIST vary across differently sized cell types?

In the methods section, the authors write they removed 49 of 88 TMA cores from their dataset due to imaging artifacts, tissue loss, registration failure, or non-consecutive tissue sections so as to evaluate COEXIST only on very high-quality data. However, implementation of this algorithm by others will necessitate its application on non-ideal datasets. The authors should analyze a small number of non-perfect serial images (non-consecutive, imperfectly registered, etc.) to quantify the loss in accuracy compared to higher quality datasets. Alternatively, the authors could use their simulation framework to iteratively reduce the quality of their virtual dataset and compute the loss in accuracy here. In the discussion section, the authors should provide further detail regarding the technical limitations of this software in non-perfect serial datasets.

References:

[1] Geraud et al, “Three-dimensional computer reconstructions from serial sections of cell nuclei”, Biology of the Cell, 1988

Reviewer #2: The authors present a novel method for linking molecular profiles across slices in multiplexed tissue imaging, addressing the challenge of incomplete molecular profiles and potential biases in cell type phenotyping due to tissue sectioning and low mRNA counts. While previous approaches used molecular profiles to match cells across slices via a linear assignment problem, the authors compellingly argue that such methods can artificially reduce molecular profile diversity by enforcing cell consistency. Instead, their proposed COEXIST method links cells by minimizing the 1-Spearman correlation coefficient of overlapping cells' molecular profiles, thereby removing the need for predefined maximum distances or anchors. The approach is validated through a didactic Monte-Carlo simulation and tested on several publicly available datasets, which is commendable.

I had some difficulty fully grasping the COEXIST method solely from the manuscript without referring to the provided code. Specifically, the integration of spatial localization and molecular profiles, which draws upon concepts from cell tracking and linear sum assignment discussed in preceding sections of the Results, could be further clarified. The method's presentation, while concise, leaves a few points ambiguous. I recommend briefly reiterating the key steps or including an algorithm pseudocode to enhance clarity. For instance, on page 20, the statement "We use D as described previously in place of the distance measure (region overlap or centroid distance) in the cell tracking algorithm. Therefore, spatially coherent candidate cells are considered, but matches are assigned by shared molecular profiles" might be less confusing if rephrased. Perhaps something like: "Across a set of spatially overlapping cells, matching cells are defined by minimizing the 1-Spearman correlation coefficient D between their molecular profiles," assuming this interpretation is correct.

On page 8, the authors note that 'LSA, ... can yield unstable results depending on the order of reference.' Could the authors elaborate on whether the approach would exhibit order invariance if the cost matrix were symmetric?

Page 17 mentions the removal of nonconsecutive slices during quality control. It would be valuable to understand the sensitivity of the proposed approach to registration errors between consecutive slices. Inaccuracies in cell registration could potentially invalidate the tracking approach. I suggest the authors consider adding an experiment to investigate the impact of such registration errors.

The cell segmentation on page 17 is performed using the mesmer cell segmentation method [45]. Given that mesmer can also perform 3D segmentation by linking regions across sections, it would be beneficial if the authors could discuss this alternative approach within the text.

On page 18, I was unable to identify the principle of Voronoi partitioning via stable matching [36] in the described tracking approach. The current method appears to match cells based on the maximum overlap between nuclear masks, which seems unrelated to algorithm (1) in reference [36]. As there is no apparent Voronoi partitioning or stable matching algorithm employed in the text or code, I suggest reviewing or removing this reference.

Regarding the notation on page 18, in 'P ∩ Q defined as subsets P ⊆ P and Q ⊆ Q', it appears that P and Q are subsets of themselves by definition. Could the authors clarify this notation?

Also on page 18, could the authors please clarify the meaning of the notation n(P)=n(Q)?

Finally, on page 21, the decision to perform clustering after cell matching is highly relevant and effectively showcases the utility of the cell matching. However, it's not entirely clear why clusters are computed solely on cells shared across slices, rather than considering all cells for cluster definition. Further clarification on this choice would be appreciated.

**Have the authors made all data and (if applicable) computational code underlying the findings in their manuscript fully available?**

Reviewer #1: Yes

Reviewer #2: Yes

PLOS authors have the option to publish the peer review history of their article (what does this mean? ). If published, this will include your full peer review and any attached files.

**Do you want your identity to be public for this peer review?** For information about this choice, including consent withdrawal, please see our Privacy Policy .

Reviewer #1: No

Reviewer #2: **Yes: ** Jérôme Boulanger

**Figure resubmission:**

**Reproducibility:**



---

## [Decision Letter · Decision Letter 1]

11 Jul 2025

Dear Dr. Chang,

We are pleased to inform you that your manuscript 'COEXIST: Coordinated single-cell integration of serial multiplexed tissue images' has been provisionally accepted for publication in PLOS Computational Biology.

Best regards,

Marc Birtwistle

Section Editor

PLOS Computational Biology

Reviewer's Responses to Questions

**Comments to the Authors:**

Reviewer #1: I am satisfied with the authors response to my previous comments and have no recommendations warranting further revision. I recommend acceptance of this manuscript for publication.

Reviewer #2: I'd like to thank the authors for their diligent work in incorporating the reviewers' feedback. The enhanced clarity of the COEXIST algorithm greatly improves its accessibility and understanding. Furthermore, the newly added insights into the sensitivity to registration errors are highly beneficial, offering a more complete picture of the approach's applicability and constraints.

**Have the authors made all data and (if applicable) computational code underlying the findings in their manuscript fully available?**

Reviewer #1: Yes

Reviewer #2: Yes

PLOS authors have the option to publish the peer review history of their article (what does this mean? ). If published, this will include your full peer review and any attached files.

**Do you want your identity to be public for this peer review?** For information about this choice, including consent withdrawal, please see our Privacy Policy .

Reviewer #1: **Yes: ** Ashley Kiemen

Reviewer #2: **Yes: ** Jérôme Boulanger

---

## [Editor Report · Acceptance letter]

PCOMPBIOL-D-25-00713R1

COEXIST: Coordinated single-cell integration of serial multiplexed tissue images

Dear Dr Chang,

I am pleased to inform you that your manuscript has been formally accepted for publication in PLOS Computational Biology. Your manuscript is now with our production department and you will be notified of the publication date in due course.

With kind regards,

Anita Estes
